# Reduced meiotic recombination in rhesus macaques and the origin of the human recombination landscape

Cheng Xue[1¤]*, Navin Rustagi[1], Xiaoming Liu[2], Muthuswamy Raveendran[1], R. Alan Harris[1], Manjunath Gorentla Venkata[3], Jeffrey Rogers[1]*, Fuli Yu[1]*

1 Human Genome Sequencing Center and Department of Molecular and Human Genetics, Baylor College of Medicine, Houston, Texas, United States of America, 2 USF Genomics & College of Public Health, University of South Florida, Tampa, Florida, United States of America, 3 Oak Ridge National Laboratory, Oak Ridge, Tennessee, United States of America

¤ Current address: Genovo Inc., San Diego, California, United States of America
* fyu@bcm.edu (FY); jr13@bcm.edu (JR); cheng.xue.2013@gmail.com (CX)

**Data Availability Statement:** All relevant data are within the paper and its Supporting Information files.

## Abstract

Characterizing meiotic recombination rates across the genomes of nonhuman primates is important for understanding the genetics of primate populations, performing genetic analyses of phenotypic variation and reconstructing the evolution of human recombination. Rhesus macaques (*Macaca mulatta*) are the most widely used nonhuman primates in biomedical research. We constructed a high-resolution genetic map of the rhesus genome based on whole genome sequence data from Indian-origin rhesus macaques. The genetic markers used were approximately 18 million SNPs, with marker density 6.93 per kb across the autosomes. We report that the genome-wide recombination rate in rhesus macaques is significantly lower than rates observed in apes or humans, while the distribution of recombination across the macaque genome is more uniform. These observations provide new comparative information regarding the evolution of recombination in primates.

## Introduction

Meiotic recombination is a fundamental genetic process that serves essential cellular functions [1, 2]. The frequency and distribution of recombination events shape the structure of haplotypes within a population, and thus influence population-level response to positive selection. This ultimately affects the distribution of phenotypic variation across environments [3, 4]. Consequently the structure of recombination maps and patterns of linkage disequilibrium in humans and other organisms are critical parameters in efforts to map genes that affect disease or reconstruct population genetic processes that shape the genomic and phenotypic diversity within species [5, 6].

In the human genome, the amount and pattern of recombination has been investigated in various ways. Researchers have quantified recombination through linkage mapping in large pedigrees [7, 8], and generated fine-scale recombination maps using single nucleotide

**Funding:** This work was supported by National Institutes of Health (NIH) grants R24-OD011173 to J.R. and 5R01HG008115 to F.Y.

**Competing interests:** The authors have declared that no competing interests exist.

polymorphisms (SNPs) genotyped in unrelated individuals [9, 10]. Small genomic segments (<2–3 kb) that exhibit much higher rates of recombination than the genome-wide average (so-called recombination hotspots) account for most recombination events [8] and evolve quickly [11, 12]. Human recombination rates differ within and among populations [13] while humans and chimpanzees (*Pan troglodytes*) share only a small proportion of all their hotspots [11]. Hotspots also evolve rapidly in other species [14]. Amino acid changes in the protein PRDM9 are associated with evolutionary changes in the location of recombination hotspots [15].

While the specific sites of recombination evolve rapidly, the overall rate of recombination is quite similar among humans, chimpanzees and other great apes. Stevison et al. [12] analyzed chimpanzees (*Pan troglodytes*), bonobos (*Pan paniscus*) and gorillas (*Gorilla gorilla*), reporting that particular recombination hotspots are gained and lost rapidly in that clade, but that the genome-wide rate of recombination is not remarkably different between species. To provide a broader comparative context and new information regarding a widely used laboratory primate species, we investigated fine-scale recombination in rhesus macaques (*Macaca mulatta*). As Old World monkeys, rhesus macaques share a last common ancestor with humans and other hominoids approximately 25–28 million years ago [16]. We previously estimated the amount of DNA polymorphism in rhesus macaques [17] and generated an initial fine-scale recombination map for this species, using low coverage sequence data from 49 individuals. In the present analysis, we report a more complete analysis of a high-resolution recombination map based on higher coverage whole genome sequencing of 123 Indian-origin rhesus macaques. The dataset consists of ~18 million common (minor allele frequency > 0.05) SNPs and provides new perspective on the evolution and distribution of recombination hotspots in *Homo sapiens* and other primates.

## Methods and materials

We used whole genome sequence data from 123 Indian-origin rhesus macaques to develop the new rhesus recombination map. DNA from these macaques was used to create standard Illumina paired-end sequencing libraries. After passing library quality control, the libraries were sequenced using the Illumina HiSeq 2000 platform, following manufacturer's recommended procedures. The average genome-wide read depth across samples was 26.7x, ranging from 7.0x to 60.7x. Single nucleotide polymorphism (SNP) genotypes were generated by first mapping the quality-filtered sequence reads to the rheMac2 rhesus macaque reference assembly using BWA [18], and then using SNPtools [19] to identify high quality variants. These data were originally used in the population genetic analyses reported by Xue et al. [17].

We used *LDhat* software version 2.1 (download from https://github.com/auton1/LDhat) to phase the rhesus macaque haplotypes and compute the fine-scale genetic map for these animals (n = 123). Computations were performed using the supercomputer TITAN at Oak Ridge National Laboratory. To assess our analysis pipeline, we downloaded 1000 Genomes OMNI haplotype data (ftp://ftp.1000genomes.ebi.ac.uk/vol1/ftp/phase1/analysis_results/supporting/omni_haplotypes) and used 1000 Genomes OMNI genetic map (downloaded from ftp://ftp.1000genomes.ebi.ac.uk/vol1/ftp/technical/working/20130507_omni_recombination_rates/) as our standard for testing. We independently produced genetic maps for YRI (n = 100), CEU (n = 99) and CHB (n = 98).

To begin, we used the *LDhat* program *complete* to generate a lookup table for 100 human (YRI) samples with $\theta = 0.001$ and another lookup table for 123 Indian rhesus samples with $\theta = 0.0025$ [17]. Next, we used the program *lkgen* to generate the lookup tables for CEU (n = 99) and CHB (n = 98) samples. We then ran *interval* to estimate recombination rates. For further

details, see Xue et al. [17] and Auton and McVean [20]. We next divided the autosomal haplotype data into windows including 4000 SNPs each, with overlap of 200 SNPs between adjacent windows. We performed 60 million iterations with block penalty of 5 and first burn-in iterations (20 million) discarded. Finally, the genetic maps were combined across adjacent (overlapping) windows as previously described [11, 20]. We removed chromosomal regions with unusual patterns of linkage disequilibrium using filters from [11]. If $4N_er$ estimated between two adjacent SNPs is >100, or a gap within a window is >50kb, we set the recombination rate to zero for the surrounding ±50 SNPs upstream and downstream. To obtain recombination rates for autosomes, we split autosomes into 1Mb, 100kb and 10kb bins. For each bin, we interpolated the genetic distance using the cumulative genetic map. The bins in the first 5Mb of telomeric regions, and the 5Mb regions adjacent to the centromeres were excluded. The genetic distance was calculated for each bin by interpolation. The bins in which the proportion of gaps in the assembly reference genome ("N"s, unclear nucleotide in the reference genome) was >0.1 or the proportion of SNPs zeroed >0.1, were also discarded.

Comparative data regarding recombination rates in chimpanzees were obtained from Auton et al. [11] and Winckler et al. [21].

## Results

We estimated autosomal recombination rates across 123 Indian-origin rhesus macaques using LDhat version 2.1 [20, 22] and phased genotypes obtained from whole genome sequencing. To estimate the rhesus fine-structure recombination map, we first tuned and validated our analytical pipeline by replicating a previous study, producing a fine-scale genetic map of the human genome using 1000 Genomes OMNI haplotype data (The 1000 Genomes Project Consortium, ftp://ftp.1000genomes.ebi.ac.uk//vol1/ftp/phase1/analysis_results/supporting/omni_haplotypes). We used the 1000 Genomes OMNI genetic map (downloaded from ftp://ftp.1000genomes.ebi.ac.uk/vol1/ftp/technical/working/20130507_omni_recombination_rates/) as a standard against which to compare our results. Employing our pipeline to re-calculate recombination across each human autosome, the correlation coefficient $r^2$ between our results and the 1000 Genomes results was ~1.0 (Fig 1A). The distance in cM for a given physical DNA sequence length, and the calculated $r^2$ for each scaled window comparing our estimates and the 1000 Genomes OMNI maps for Yoruban (YRI), Chinese (CHB) and European (CEU) samples also show strong correlations ($r^2$ is ~0.9, Fig 1B). We therefore used the same pipeline in generating the new rhesus macaque genetic map.

We constructed our recombination map for macaques using 18 million SNPs genotyped in 123 unrelated Indian-origin rhesus macaques, part of the dataset previously analyzed by Xue et al. [17] (see S1 Table in S1 File for list of samples and sample providers). The recombination map was calculated employing the same parameters used by Xue et al. [17] for a smaller scale analysis, including $4N_e\mu = 0.0025$. Fig 2 illustrates our new results for 1 Mb windows across all 20 macaque autosomes. The calculated macaque genome-wide average recombination rate is 0.448 + 0.286 cM/Mb and the population-scaled rate is $\rho = 0.112 + 0.07$, where $\rho = 4 \cdot N_e \cdot r$ and r is the recombination rate per generation per kilobase. Like the human genome, recombination in macaques varies locally along chromosomes and is more frequent in regions near telomeres. The complete autosomal map can be viewed in the context of either the rheMac2 or rheMac8 rhesus genome reference assemblies by accessing this URL (https://www.hgsc.bcm.edu/non-human-primates/rhesus-monkey-genome-project) and clicking on the links for UCSC track hubs. These browser views allow one to search for any specific region within the rhesus genome and see the details of the fine-structure variation in local recombination rates.

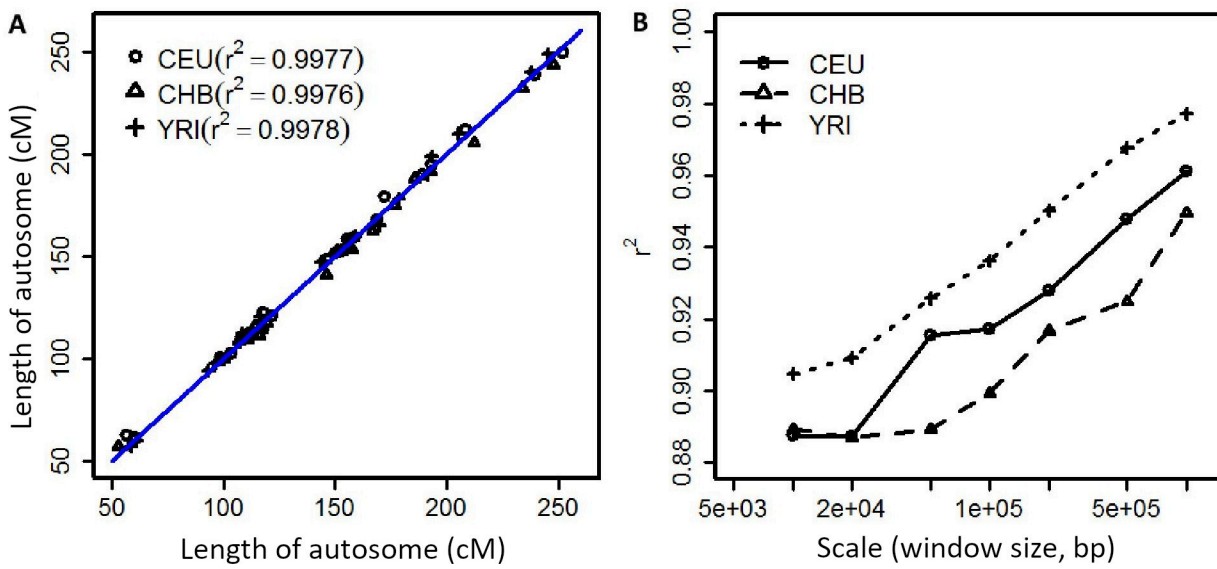

**Fig 1. Quality control of LDHat pipeline using 1000 Genomes OMNI data.** Subplot A. The relationship between the length of autosomes in unit of cM calculated by our pipeline and the standard used for evaluation (downloaded from ftp://ftp.1000genomes.ebi.ac.uk/vol1/ftp/technical/ working/20130507_omni_recombination_rates/). Each point is a length of an autosome. Subplot B. Correlation coefficients between recombination rates we observed and those from the 1000 Genomes with different window scales.

We also found that linkage disequilibrium decays to less than $r^2 = 0.20$ at about 300 kb in the macaque genome (S1 Fig in S1 File).

The genome-wide recombination rate for macaques is remarkably lower than the values published for humans and chimpanzees [11, 12]. Fig 3 illustrates the distribution of rates across the three species using 1 Mb, 100 kb and 10 kb windows. The distributions are quite different in the macaques, regardless of window size used. In our re-analysis of human data, the calculated rate for Africans (YRI) is 1.092 ± 0.791 cM/Mb, for Europeans is 1.093 ± 0.791 cM/ Mb and for Chinese is 1.094 ± 0.827 cM/Mb. Both the mean and variance in recombination rates are higher in humans than in rhesus macaques (F-test for differences in variances, $p < 2.2 \times 10^{-16}$ comparing rhesus with any of the three human populations).

As in prior studies [10, 11], the shapes of the distributions of recombination rates for the human genome differ among the three window sizes. Using 1 Mb windows the human distribution has a peak window count (density) about 0.5–0.7 cM/Mb. Using 10kb windows, the human peak density is at the lowest recombination rates. In contrast, the shape of the three rate distributions for rhesus macaques are all similar, sharing a peak about 0.2–0.3 cM/Mb. Across all window sizes humans and chimpanzees have higher proportions of their genomes exhibiting high recombination rates (>1.5 cM/Mb) than do macaques. Humans also display a higher proportion of the genome with very low (<0.1 cM/Mb) recombination. Gini coefficients show the macaques have a more even distribution of recombination rates across the genome compared with chimpanzee and human. As a result of these differences in the pattern of recombination rates across the human and rhesus macaque genomes, the correlation of local rates is low. S2 Fig in S1 File presents the correlations of rates in orthologous genomic segments using various window sizes. The similarities between the macaques and humans are low throughout the range of analyses.

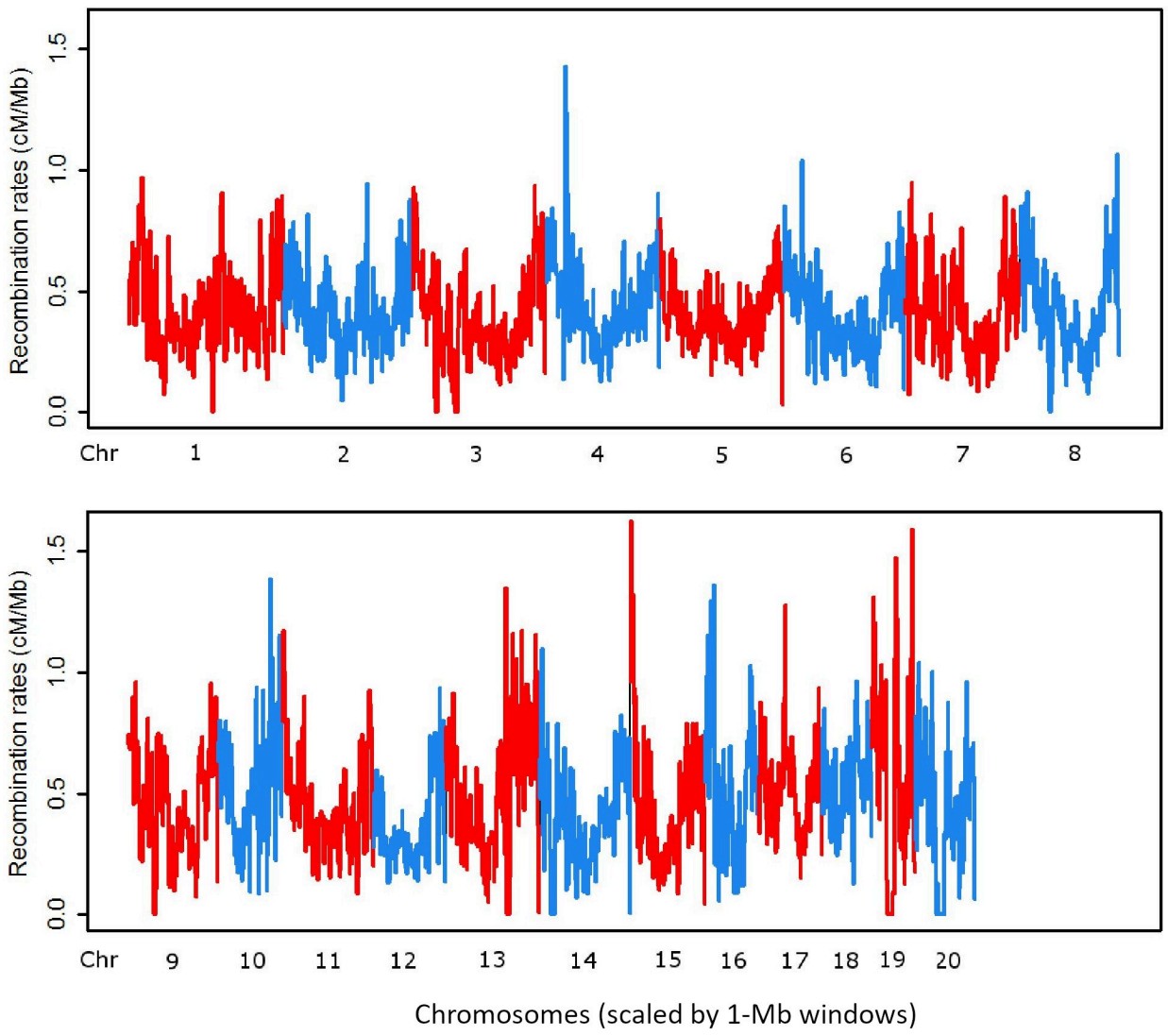

**Fig 2. Autosomal landscape of recombination rates for 1Mb-window scale in the Indian-origin rhesus macaque genome.**

## Discussion

Rates of recombination per megabase are known to differ across vertebrate species. The estimated rate in laboratory mice is substantially lower (~0.5 cM/Mb) than in humans, while other mammals display intermediate values [1]. The great apes share relatively high average rates of recombination with humans [12]. Our results indicate rhesus macaques display a notably lower genome-wide recombination rate relative to humans, consistent with earlier less detailed studies. Employing pedigree-based linkage analysis of microsatellites in the macaques [23] and in *Papio* baboons [24], we reported low rates of recombination. In addition, the correlations between local recombination rates when comparing orthologous genomic segments in humans and macaques are low.

However, it is the distribution of local recombination rates across the genome that is the most unexpected finding of this study. Rhesus macaques display a unimodal distribution with the largest density of genomic windows (regardless of window size used) showing 0.2–0.3 cM/Mb. In contrast, the shape of the distribution of recombination rates across the human genome

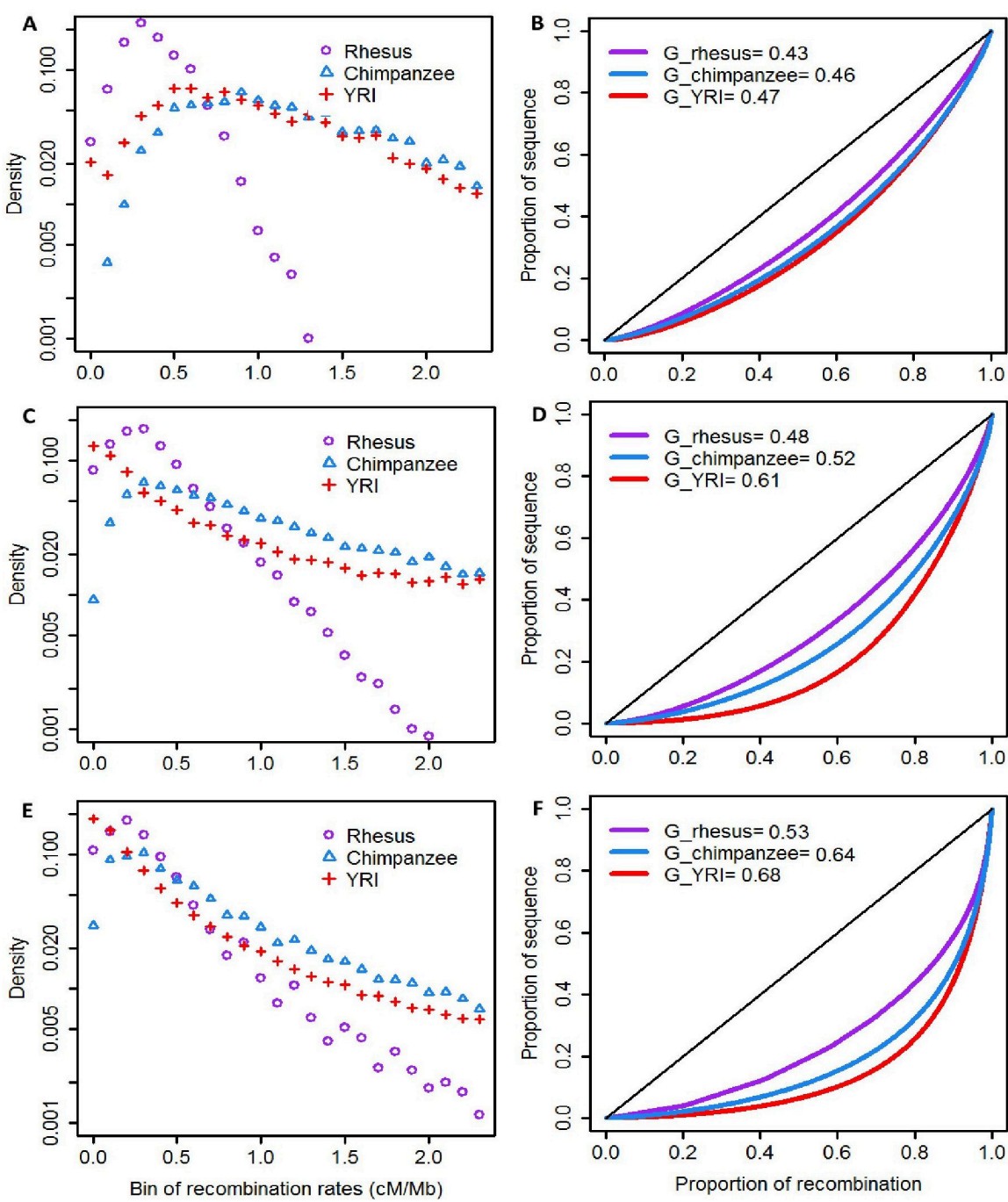

**Fig 3. Distribution of autosomal recombination rates for different scaled windows in rhesus, chimpanzee and human (YRI) genomes.** Subplots A, C and E are density of recombination rates with 1-Mb, 100Kb and 10Kb windows; subplots B, D and F present Lorenz curves for the distributions of recombination rates using 1-Mb, 100Kb and 10Kb windows. "G_rhesus", "G_chimpanzee" and "G_YRI" are Gini coefficients for rhesus, chimpanzee and human (YRI), respectively.

depends greatly on the window size employed. Large (1 Mb) regions show peak density between 0.5–1.0 cM/Mb while smaller windows exhibit much lower modal rates (<0.1 cM/Mb). Furthermore, the humans and chimpanzees both show many windows with rates >1.5 cM/Mb across all window sizes.

The explanation for this inter-species difference of genome-wide recombination rate is not readily apparent due to different effective population size and demographics [25–28]. Munch et al. [29] showed that the human recombination map has changed more than the chimpanzee map since these two lineages diverged about 7 million years ago. We infer that the common ancestor of humans and chimpanzees had a recombination rate higher than that in rhesus macaques, possibly due to a large number of 100kb and 10kb regions with rates >1.5 cM/Mb. Since the divergence of humans and chimpanzees the human genome has apparently reduced recombination hotspots in many genomic regions (or reduced the rate of recombination in those hotspots), and consequently the human genome has more 100 kb windows with very low recombination (<0.2 cM/Mb), relative to chimpanzees. This increase in low recombination segments has not significantly reduced the overall human rate of recombination. These findings suggest that the extreme intra-genome, inter-region variance in human recombination rates, exhibiting twice the standard deviation estimated for macaques, may be a recent phenomenon. The evolutionary history shared by humans and chimpanzees after the divergence of Old World monkeys apparently included an increase in the fraction of the genome showing high rates of recombination (>1.5 cM/Mb).

In primates, recombination events generally occur within hotspot locations that are regulated by PRDM9 [10, 30, 31]. The zinc finger domain of PRDM9 binds DNA sequences and initiates double strand breaks that generate recombination events [15, 30]. Myers et al. [30] compared PRDM9 zinc finger domains across seven mammalian species, finding the number of amino acids in PRDM9 zinc finger domains in rhesus macaques is smaller than chimpanzee and human, which might result in shorter DNA binding sites in rhesus. If macaque PRDM9 binds to more genomic sites than human PRDM9, this may produce a more even distribution of recombination rates across that genome compared with humans (Fig 3).

On the other hand, the PRDM9 zinc finger domain and its binding sites coevolve rapidly [31, 32] and the gain or loss of "hot" and "cold" spots dictates the specific location of recombination events [30, 33]. Thus, our observation of a consistent distribution of recombination rates in rhesus macaques and window-size specific distribution in humans is consistent with previous studies of PRDM9 differences across primates. Higher effective population size of rhesus macaques [17] may also be a factor if hotspots appear and disappear rapidly relative to the evolutionary coalescence of individual haplotypes. Additional studies describing recombination maps across more primate species will help define the evolutionary history of recombination rates and their intra-genome distributions, and will allow us to determine whether the pattern observed in hominoid genomes is unique among primates.

## Supporting information

**S1 File.**
(DOCX)

## Acknowledgments

We thank Donna Muzny, Richard Gibbs and the staff of the Baylor College of Medicine Human Genome Sequencing Center for essential contributions to the production of the macaque SNP data. We are indebted to the following primate research centers for providing blood and/or DNA samples from the study macaques: California National Primate Research Center (NPRC), New England NPRC, Oregon NPRC, Southwest NPRC, Tulane NPRC, Wisconsin NPRC, Yerkes NPRC and the Caribbean Primate Research Center. We also thank Oak Ridge National Laboratory (ORNL) for providing their high-performance supercomputer

resources for the analyses in this study. Kasper Munch provided valuable suggestions related to rhesus macaque effective population size.

## Author Contributions

**Conceptualization:** Xiaoming Liu, Fuli Yu.

**Data curation:** Cheng Xue, Muthuswamy Raveendran, R. Alan Harris.

**Formal analysis:** Cheng Xue, Navin Rustagi, R. Alan Harris.

**Funding acquisition:** Fuli Yu.

**Investigation:** Cheng Xue, Navin Rustagi, Fuli Yu.

**Methodology:** Cheng Xue.

**Project administration:** Jeffrey Rogers.

**Resources:** Muthuswamy Raveendran, Manjunath Gorentla Venkata.

**Software:** Manjunath Gorentla Venkata.

**Supervision:** Jeffrey Rogers, Fuli Yu.

**Validation:** Cheng Xue.

**Visualization:** Cheng Xue.

**Writing – original draft:** Cheng Xue, Jeffrey Rogers.

**Writing – review & editing:** R. Alan Harris, Jeffrey Rogers, Fuli Yu.

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
