## [Decision Letter · Decision Letter 0]

9 Jun 2020

PONE-D-20-16003

Reduced meiotic recombination in rhesus macaques and and the origin of the human recombination landscape

PLOS ONE

Dear Dr. Xue,

Thank you for submitting your manuscript to PLOS ONE. After careful consideration, we feel that it has merit but does not fully meet PLOS ONE’s publication criteria as it currently stands. Therefore, we invite you to submit a revised version of the manuscript that addresses the points raised during the review process.

We look forward to receiving your revised manuscript.

Kind regards,

Karol Sestak

Academic Editor

PLOS ONE

Journal Requirements:

We thank Donna Muzny, Richard Gibbs and the staff of the Baylor College of Medicine

Human Genome Sequencing Center for essential contributions to the production of the

macaque SNP data. We are indebted to the following primate research centers for

providing blood and/or DNA samples from the study macaques: California National

Primate Research Center (NPRC), New England NPRC, Oregon NPRC, Southwest

NPRC, Tulane NPRC, Wisconsin NPRC, Yerkes NPRC and the Caribbean Primate

Research Center. We also thank Oak Ridge National Laboratory (ORNL) for providing

their high-performance supercomputer resources for the analyses in this study. Kasper

Munch provided valuable suggestions related to rhesus macaque effective population

size, and two anonymous reviewers provided helpful comments, critiques and

suggestions. This work was supported by National Institutes of Health (NIH) grants

R24-OD011173 to J.R., U54-HG006484-01 to R. G. and 5R01HG008115 to F.Y.

No

No

5. Please amend the manuscript submission data (via Edit Submission) to include author Navin Rustagi, Xiaoming Liu, Muthuswamy Raveendran, R. Alan Harris, Manjunath Gorentla Venkata.

6. Please upload a copy of Supporting Information Table 1. which you refer to in your text on page 13.

Reviewers' comments:

Reviewer's Responses to Questions

**Comments to the Author**

1. Is the manuscript technically sound, and do the data support the conclusions?

Reviewer #1: Partly

Reviewer #2: Yes

2. Has the statistical analysis been performed appropriately and rigorously? 

Reviewer #1: Yes

Reviewer #2: Yes

3. Have the authors made all data underlying the findings in their manuscript fully available?

Reviewer #1: Yes

Reviewer #2: Yes

4. Is the manuscript presented in an intelligible fashion and written in standard English?

Reviewer #1: Yes

Reviewer #2: Yes

5. Review Comments to the Author

Reviewer #1: In this paper, Xue and colleagues infer a fine-scale recombination map in rhesus macaques by applying LDhat to 123 sequenced macaques. I find the results interesting and the genetic map will be useful for comparative studies but I am not convinced by the main result that there is reduced overall recombination in macaques.

Major comments:

One of the main results of the paper is that macaques have a lower overall recombination rate than other great apes. Because LDhat infers a population-scaled recombination rate, it can be difficult to disentangle differences in effective population size from differences in true recombination rate. The scaling gets even murkier when populations have undergone size changes (see Johnston and Cutler AJHG 2012; Kamm et al. Genetics 2016; Dapper and Payseur MBE 2017; and Spence and Song 2019). As a result, I would not put too much stock in the absolute rates of recombination. Relative rates, across the genome within a single species, are somewhat more robust to differences in demographic history, so I find the correlation results to be much more convincing.

Figure S1 seems weird to me. I would expect r^2 to asymptote to something close to 1 / 2n, so 1 / 236, which is approximately 0.004, but it looks like r^2 is actually asymptoting to about 0.15, which seems very high. If this is not an error, it could suggest large amounts of population structure in the sample (leading to so-called "ancestry LD" at long ranges). Large amounts of population structure violate the panmixia assumptions of LDhat and could lead to downward biased recombination rate estimates. One way to assess population structure would be to visualize the genotype data with PCA or ADMIXTURE (Alexander, Novembre, and Lange, Genome Research, 2009) and see if the data show distinct clusters, or spread out along a cline.

I find Figure S2 to be very interesting and matches the overall picture that has emerged that at very fine-scales recombination maps do not correlate well across great ape species, but they do at broader scales. This suggests a slow evolving broad-scale mechanism of recombination patterning, but a fast-evolving fine-scale mechanism (i.e., PRDM9).

Minor comment:

LDhat does not phase data but in the manuscript it says "We used LDhat software version 2.1 ... to phase the rhesus macaque haplotypes...". Were the data phased? If so, how? If the data were not phased, my experience has been the LDhat can produce downward biased recombination estimates, which would be consistent with the findings here.

Reviewer #2: Well written manuscript with interesting data. There are few minor recommendation for improvement:

1) English grammar and symbol use should be improved. For example, Figures 1 and 2 (axes X and Y) are not properly labeled. Figures and their legends including coloration of these figures should be improved for the sake of clarity.

6. PLOS authors have the option to publish the peer review history of their article (what does this mean?). If published, this will include your full peer review and any attached files.

Reviewer #1: No

Reviewer #2: No

---

## [Author Response · Author response to Decision Letter 0]

30 Jun 2020

Dear Dr. Karol Sestak,

 We have revised our manuscript entitled “Reduced meiotic recombination in rhesus macaques and the origin of the human recombination landscape” in order to address comments from you and the reviewers. Please see our responses to these comments included below. 

 Thank you for consideration.

 Best regards

 Fuli Yu, Jeffery Rogers and Cheng Xue

Response: Thank you for reminding us of this. We revised the manuscript accordingly. 

We thank Donna Muzny, Richard Gibbs and the staff of the Baylor College of Medicine

Human Genome Sequencing Center for essential contributions to the production of the

macaque SNP data. We are indebted to the following primate research centers for

providing blood and/or DNA samples from the study macaques: California National

Primate Research Center (NPRC), New England NPRC, Oregon NPRC, Southwest

NPRC, Tulane NPRC, Wisconsin NPRC, Yerkes NPRC and the Caribbean Primate

Research Center. We also thank Oak Ridge National Laboratory (ORNL) for providing

their high-performance supercomputer resources for the analyses in this study. Kasper

Munch provided valuable suggestions related to rhesus macaque effective population

size, and two anonymous reviewers provided helpful comments, critiques and

suggestions. This work was supported by National Institutes of Health (NIH) grants

R24-OD011173 to J.R., U54-HG006484-01 to R. G. and 5R01HG008115 to F.Y.

No

Response: We removed the funding-related text from the manuscript, and updated the Funding Statement in the submission.

No

Response: We included the statement “The authors have declared that no competing interests exist.” in the cover letter and uploaded it to the submission site. We also updated the Competing Interests section in the submission.

Response: Thanks for reminding us. We revised the title on the online submission.

5. Please amend the manuscript submission data (via Edit Submission) to include author Navin Rustagi, Xiaoming Liu, Muthuswamy Raveendran, R. Alan Harris, Manjunath Gorentla Venkata.

 Response: We added all authors in the online submission.

6. Please upload a copy of Supporting Information Table 1. which you refer to in your text on page 13.

Response: We uploaded this table.

5. Review Comments to the Author

Reviewer #1: In this paper, Xue and colleagues infer a fine-scale recombination map in rhesus macaques by applying LDhat to 123 sequenced macaques. I find the results interesting and the genetic map will be useful for comparative studies but I am not convinced by the main result that there is reduced overall recombination in macaques.

Major comments:

One of the main results of the paper is that macaques have a lower overall recombination rate than other great apes. Because LDhat infers a population-scaled recombination rate, it can be difficult to disentangle differences in effective population size from differences in true recombination rate. The scaling gets even murkier when populations have undergone size changes (see Johnston and Cutler AJHG 2012; Kamm et al. Genetics 2016; Dapper and Payseur MBE 2017; and Spence and Song 2019). As a result, I would not put too much stock in the absolute rates of recombination. Relative rates, across the genome within a single species, are somewhat more robust to differences in demographic history, so I find the correlation results to be much more convincing.

Response: We thank the Reviewer for the constructive suggestion. We revised and added text in the Discussion, “The explanation for this inter-species difference of genome-wide recombination rate is not readily apparent due to different effective population size and demographics”. 

Figure S1 seems weird to me. I would expect r^2 to asymptote to something close to 1 / 2n, so 1 / 236, which is approximately 0.004, but it looks like r^2 is actually asymptoting to about 0.15, which seems very high. If this is not an error, it could suggest large amounts of population structure in the sample (leading to so-called "ancestry LD" at long ranges). Large amounts of population structure violate the panmixia assumptions of LDhat and could lead to downward biased recombination rate estimates. One way to assess population structure would be to visualize the genotype data with PCA or ADMIXTURE (Alexander, Novembre, and Lange, Genome Research, 2009) and see if the data show distinct clusters, or spread out along a cline.

Response: We did PCA analysis on the genotype data (Please see figure below). Samples from the National Primate Research Centers form a cline with no identifiable population structure. The Cayo Santiago animals from the Caribbean Primate Research Center (green color) do form a cluster based on PC1. The PC1 eigenvalue is 2.8. This accounts for 46.91% of the variance seen across the samples using a CNG scree test that retained the top 3 principal components.

I find Figure S2 to be very interesting and matches the overall picture that has emerged that at very fine-scales recombination maps do not correlate well across great ape species, but they do at broader scales. This suggests a slow evolving broad-scale mechanism of recombination patterning, but a fast-evolving fine-scale mechanism (i.e., PRDM9).

Minor comment:

LDhat does not phase data but in the manuscript it says "We used LDhat software version 2.1 ... to phase the rhesus macaque haplotypes...". Were the data phased? If so, how? If the data were not phased, my experience has been the LDhat can produce downward biased recombination estimates, which would be consistent with the findings here.

Response: The input data of LDhat is the phased data (Xue et al. 2016). The phased data can be accessed here ftp://ftp.hgsc.bcm.edu/ucscHub/rhesusSNVs/rheMac2/Genotype_biallelic_SNP_autosomes_rhesus_V2.0_Mar5-2015.for.hub.vcf.gz. 

Reviewer #2: Well written manuscript with interesting data. There are few minor recommendation for improvement:

1) English grammar and symbol use should be improved. For example, Figures 1 and 2 (axes X and Y) are not properly labeled. Figures and their legends including coloration of these figures should be improved for the sake of clarity.

Response: We revised the figures’ labels and improved the resolution of figures.

---

## [Editor Report · Decision Letter 1]

6 Jul 2020

Reduced meiotic recombination in rhesus macaques and the origin of the human recombination landscape

PONE-D-20-16003R1

Dear Dr. Xue,

We’re pleased to inform you that your manuscript has been judged scientifically suitable for publication and will be formally accepted for publication once it meets all outstanding technical requirements.

Kind regards,

Karol Sestak

Academic Editor

PLOS ONE

---

## [Editor Report · Acceptance letter]

11 Aug 2020

PONE-D-20-16003R1 

Reduced meiotic recombination in rhesus macaques and the origin of the human recombination landscape 

Dear Dr. Xue:

I'm pleased to inform you that your manuscript has been deemed suitable for publication in PLOS ONE. Congratulations! Your manuscript is now with our production department. 

Kind regards, 

on behalf of

Dr. Karol Sestak 

Academic Editor

PLOS ONE